# Correlation between quality of life of cardiac patients and caregiver burden

**Maha Subih**[ID]*, **Marwa AlBarmawi**, **Dalal Yehia Bashir**, **Shrooq Munir Jacoub**, **Najah Sayyah Sayyah**

School of Nursing-Al-Zaytoonah University of Jordan (ZUJ), Amman, Jordan

☯ These authors contributed equally to this work.
* maha.subih@zuj.edu.jo

## Abstract

### Background

Caregivers experience high strain related to care giving. There is increasing interest in examining the caregiver burden of cardiac patients and studying the characteristics of caregivers.

### Purpose

To explore the correlation between quality of life cardiac patients and caregiver's burden.

### Methods

A cross-sectional design using a convenience sample of caregivers and patients with cardiac conditions. Sociodemographic sheet, Dutch Objective Burden Inventory (DOBI), and Quality of Life (QLI-Cardiac 4). Linear regression was used to explore the predictors.

### Results

200 caregivers and 200 patients with cardiac diseases completed the study. The overall mean scores of both DOBI and QLI-4 indicated moderate results 1.51(SD 0.4), 19.8 (SD 4.7) respectively. Predictors of caregiver burden were young, less educated caregivers and high QoL of cardiac patients.

### Conclusion

Caregivers should receive more support and training from healthcare providers to develop their coping and resilience skills in a way that decreases their care burden and improves their quality of care and self-confidence.

## Introduction

Cardiac diseases are increasingly becoming chronic conditions that may require lifelong care [1]. Cardiovascular diseases (CVDs) are the number one cause of death globally, accounting for an estimated 31.1% of all deaths [2]. In Jordan, a developing Arabic country with an

**Data Availability Statement:** Data cannot be shared publicly because participants did not consent to share their data. Data are available from the Al Zaytoonah University of Jordan Institutional Data Access / Ethics Committee (contact via chair

of ethics committee loai.s@zuj.edu.jo) for researchers who meet the criteria for access to confidential data.

**Funding:** This work was supported by Al Zaytoonah University of Jordan (ZUJ) (Grant number 21/172/2018) The funders had no role in study design, data collection and analysis, decision to publish, or preparation of the manuscript. MS, MA, DY, SJ, NS 750$ to each one Al Zaytoonah University of Jordan: www.zuj.edu.jo

**Competing interests:** the authors have declared that no competing interest exist.

estimated population of 10.6 million including citizens and refugees, CVDs deaths reached 37% of total deaths according to the latest WHO data published in 2018 [3].

The aim of healthcare providers' care plans for cardiac patients is to "live longer and happier," rather than simply living longer [4]. It is reasonable, therefore, that health systems should be interested in the quality of life (QoL) of patients with CVD. WHO defines QoL at 1995 as the individual's perception of his or her position in life within the context of culture and value systems [5].

The presence of CVD is associated with decreased QoL among patients and increase the physical and emotional burden on caregivers [6]. Cardiac patients suffer from frequent rehospitalization either due to the disease progress or its complications during management [7]. Therefore, it has become necessary to direct attention and care to at-home caregivers, such as partners, sons, daughters, relatives, or others [8].

Despite the tradition of family members providing care to patients being highly valued in Arab and Islamic cultures as well as in other Western cultures, especially in Asian countries which have similar norms, its negative aspects influence the QoL of the caregivers [9, 10].

An individual responsible for day-to-day care and support following discharge is known as the primary caregiver [11]. The effect of providing informal care to a person with a terminal condition has been recognized as the caregiver burden [12]. It is a multidimensional response to the psychological, physical, social, emotional, and financial stressors associated with the caregiving practice [13].

Terminal illnesses lead to more responsibilities and burdens on caregivers at home or in the hospital [14]. Providing care to a seriously ill family member can compromise a caregiver's overall physical, psychosocial, and spiritual well-being [15] and lead to psychological stress and financial problems [6]. Moreover, the level of support to the patient decreases when there is any decline in the caregiver's well-being [16].

With little or no formal training, caregivers do not actively participate in discharge planning, and they may not be ready to cope or effectively manage a situation where they are expected to provide long-term systematic care and support for a chronically ill family member [17]. More structured formal training and care should be provided by health bodies, and nurse practitioners should be encouraged to help families undertake the caregiving responsibility to empower their resilience skills [18], thus improving the efficiency and quality of care provided for patients with cardiac disease as well as preserving the integrity and quality of caregivers' lives.

Locally, there is limited information regarding the prevalence of informal primary caregivers in developing countries. Additionally, there is a gap in the cardiac literature regarding the characteristics of those caregivers, which when recognized, will be useful for designing targeted training and support interventions that will guide this virtuous population gently and effectively through the whole care process.

On the other hand, QoL has been studied extensively on cardiovascular patients internationally and in Jordan specifically for heart failure (HF); however, to the researcher's knowledge, this is the first study to address this topic in Jordan. Furthermore, the mainly Western context of most existing studies means many of their findings may not be applicable in Jordanian culture; therefore this study addresses an important research gap by exploring QoL among patients with cardiac diseases and the correlation with burden on their caregivers.

## Study purpose/objectives

The purpose of this paper was to assess the QoL of cardiac patients and their caregivers' burden, and it's correlation with caregivers' burden and other demographic variables. Furthermore, this study aimed to survey the defining characteristics of primary caregivers.

## Material and methods

### Design

This study used a cross-sectional design.

### Sample and setting

According to G power (3.0.10, Germany), the minimum sample size needed was calculated to be 118 ($\alpha$ = 0.05 and power = 0.8, and medium effect size, predictors = 10). A convenience sample of 200 caregivers and 200 patients with cardiac disease agreed to participate (out of 250 questionnaires distributed giving a response rate of 80%). The care partners were identified as those who accompanied patients with cardiac diseases to the hospital clinics and identified themselves as caregivers. The respondents were chosen from one teaching hospital, two public, and three private hospital clinics in Jordan (hospitals that agreed to distribute questionnaires to patients). Inclusion criteria mandated that the caregivers and patients were older than 18 and could read and understand the Arabic language, as this was the language used in the questionnaire.

### Procedure

At the start of the study, the patients were asked if they had a family member providing care for them and, if so, whether they would like to take part in the research and comment about the care they were receiving; then the caregivers were contacted and asked if they would like to participate in the study. The participation of caregivers and patients was voluntary, with the option to withdraw at any time. The rights of confidentiality were assured, and consent forms were signed by all participants. Ethical approvals were obtained from the Institute Review Boards of both Al Zaytoonah University and the corresponding hospitals.

### Instrument

The self-administered questionnaire used in this study included a sociodemographic section and the Arabic version of the 38-item Dutch Objective Burden Inventory (DOBI) developed by Luttik, Jaarsma, Tijssen, van Veldhuisen, and Sanderman (2008), which was used to measure caregiver burden. The instrument was translated into Arabic using both forward and backward translation. The scale is divided into four domains: personal care (11 items), motivational support (10 items), emotional support (6 items), and practical and treatment-related support (11 items). Each item was rated on a Likert scale ranging from 1 (*never*) to 3 (*always*). Higher levels of caregiver burden were indicated by higher scores.

Cronbach's alpha reliability was previously found to be between 0.80 and 0.85 [19]. The English version of the DOBI had adequate and accepted internal consistency and construct validity when used in a Canadian population of caregivers [20]. This scale demonstrated good reliability in this study with a Cronbach's alpha of 0.9 and a good content validity index of 0.8. The instrument was submitted to three experts: two faculty members who hold doctorate degrees (and had many years of clinical experience in critical care units) and one clinical nurse practitioner. The experts were asked to rate each item based on relevance, clarity, and simplicity on a scale of 1 (*not relevant*) to 4 (*completely relevant*) (Polit & Beck, 2017).

The final Arabic version was piloted with 20 caregivers to verify the feasibility and the practicality of the survey after translation, and no problems were reported for any item (the pilot study was not included in the sample).

The Arabic version of QLI-Cardiac 4 was adopted to measure QoL [21]. This scale assesses satisfaction with life, using a Likert scale based on 6 points ranging between 1 (*very dissatisfied*)

and 6 (*very satisfied*). The various aspects of life were addressed by a Likert scale based on 6 points ranging from 1 (*not important to me at all*) to 6 (*very important*). The QLI-Cardiac 4 yields five scores: the overall QoL score and its four domains. The four domains of overall QoL score include psychological/spiritual, health and functioning, family, and social and economic. Five scores are generated on a 0–30 scale [21].

Cronbach's alphas have previously measured between .84 and .98 for the original QLI-Cardiac 4 (total scale)' [21]. For the Arabic version, the Cronbach's alpha internal consistency reliability coefficient for the translated version of the QLI-Cardiac 4 is .90. Higher scores on the scale indicate better QoL.

## Data analysis

Data were analyzed using the Statistical Package for Social Sciences (SPSS) version 21. The characteristics of the study sample were described using descriptive statistics (using mean (M), standard deviation (SD), percentage, and frequencies). Several correlation tests were conducted to check the relationships between caregiver burden and specific demographic variables of caregivers (Pearson, Point biserial, Eta, Biserial according to the level of measurement of the variables) then linear regression was applied to find the predictors of caregiver burden.

## Results

### Characteristics of caregivers

Two hundred caregivers completed the questionnaire. Caregivers were predominantly female, and their mean age was 40.3 (SD = 12.8). A high percentage of caregivers were married (76.5%) and were either the patient's spouse, son, or daughter (79%). Further details are given in Table 1. Two hundred cardiovascular patients were also recruited. Males represented 62.5%, with mean age 53.7(SD = 11.6) years. Most cardiac patients were diagnosed with HF (34.3%). Interestingly, most patients had a cardiac diagnosis of less than one year (44.1%); further details are presented in Table 1.

Table 1. Characteristics of the sample (caregivers: N = 200, cardiac patients: 200).

| Characteristics | Caregivers | Patients |
|---|---|---|
| | N (%) | N (%) |
| **Gender** | | |
| **Female** | 116 (58) | 124 (62.6) |
| **Male** | 84 (42) | 76 (37.4) |
| **Educational level**[*] | | |
| **Educated (more than 10 years)** | 149 (75.6) | 131 (66.5) |
| **Not educated** | 48 (24.4) | 67 (33.5) |
| **Marital status** | | |
| **Married** | 136 (68) | 153 (76.5) |
| **Unmarried** | 64 (32) | 47 (23.5) |
| **Relationship with patient** | | |
| **Spouse** | 81 (40.5) | |
| **Son/daughter** | 77 (38.5) | |
| **Sister/brother** | 24 (12) | |
| **Others** [$] | 18 (9) | |
| **Living with the patient** | | |

(*Continued*)

**Table 1.** (Continued)

| Characteristics | Caregivers | Patients |
|---|---|---|
| | **N (%)** | **N (%)** |
| **No** | 44 (22) | |
| **Yes** | 156 (78) | |
| **Age** | 40.3 (12.8) | 53.7 (11.6) |
| | 18–78 years old | 20–83 years old |
| **Employment status*** | | |
| **Employed** | 114 (57) | 96 (48.5) |
| **Unemployed** | 86 (43) | 102 (51.5) |
| **Do you deliver care to another patient?** | | |
| **No** | 170 (85) | |
| **Yes** | 30 (15) | |
| **How long have you been caring for the patient?** | | |
| **1–5 years** | 140 (70) | |
| **> 5 years** | 60 (30) | |
| **Time spent in caring (hours/day) #** | Mean (SD) | Range: 1–24 hrs |
| | 6.9 (4.6) | |
| **Duration of cardiac disease** | | |
| **Less than one year** | | 86 (44.1) |
| **1–5 years** | | 70 (35.9) |
| **> 5 years** | | 39 (20) |
| **Type of Cardiac diagnosis*** | | |
| **Myocardial Infarction (MI)** | | 26 (13.3) |
| **HF** | | 67 (34.3) |
| **PCI** | | 63 (32.3) |
| **Coronary Artery Bypass Graft (CABG)** | | 39 (20) |

*: some variables had missing data.

$: others include; parents, friends, neighbours, and relatives.

#: time spent with cardiac patients include direct and indirect care.

## Caregiver burden

The DOBI indicated that the mean score for the whole scale was 1.51 (SD = 0.4), which is considered a moderate level in comparison with other studies in the literature (no cut off point had been reported by the original authors). The highest mean scores, that is the highest subscales responsible for increasing caregiver burden, were related to the "Personal Care Burden" subscale (2.66, SD = 1.7) and "Motivational Burden" (2.3, SD 1.1) and the lowest score to the "Emotional Burden" subscale (1.51, SD 0.5), as shown in Table 2.

**Table 2. DOBI scale: Means, standard deviation, and reliability Cronbach alpha (N = 200).**

| | Mean | SD | Cronbach alpha |
|---|---|---|---|
| Total DOBI scale | 1.51 | 0.4 | 0.90 |
| Personal Care Burden | 2.66 | 1.7 | 0.88 |
| Emotional Burden | 1.51 | 0.5 | 0.83 |
| Motivational Burden | 2.30 | 1.1 | 0.87 |
| Practical/Treatment Support Burden | 1.59 | 0.5 | 0.81 |

**Table 3. Findings of the quality of life index-cardiac, version 4 (N = 200).**

| Items | Mean | SD | Range |
|---|---|---|---|
| **QoLI-Cardiac, v4** | 19.8 | 4.7 | 7.97–30 |
| **Health function** | 17.6 | 5.6 | 2.50–30 |
| **Social function** | 18.4 | 4.4 | 9.31–30 |
| **Psychological function** | 20.04 | 5.8 | 5.14–30 |
| **Family function** | 23.01 | 5.7 | 9.00–30 |

## Quality of life of cardiac patients

The overall mean score of QLI-Cardiac 4 was 19.8 (SD 4.7), which is considered moderate, and the subscales' mean scores ranged between 17.6 (SD 5.6) for the health function subscale and 23.01 (SD 5.7) for the family function (Table 3).

## Influence of caregiver characteristics on their burden

Results showed that gender, relationship with cardiac patient, and job of caregiver were significantly correlated with caregiver burden as shown in Table 4. Caregivers who were female, unemployed, with no close relationship with patients, and not living with them have a higher burden than others.

## Predictors of caregiver burden

First, caregivers' demographic variables were entered in the first step in a hierarchical regression model. Then, the QoL of cardiac patients was entered in the second step in model 2, where it was observed that the educational level ($\beta$ = -0.31. p = 0.03) and the caregiver's age ($\beta$ = -0.45. p = 0.02) were the only significant predictors of the caregiver burden; a negative sign means those caregivers that were younger and less educated were more vulnerable to a high burden than others. Overall, the demographic variables were significantly predictive ($R^2$ = 0.24, adjusted $R^2$ = 0.14, (F (9, 50) = 1.99, p = 0.03). QoL of cardiac patients was also a significant predictor ($R^2$ = 0.41, adjusted $R^2$ = 0.32, p = 0.001; Table 5), meaning that the higher the patients' QoL the higher the burden on caregivers. The overall regression including all variables was statistically significant and can predict 32% of the variance in caregiver burden.

**Table 4. Relationship between caregivers' characteristics and their burden.**

| Variables | Correlation value |
|---|---|
| | r (P value) |
| **Age** | **-0.19 (0.05)** |
| **Gender** | **0.16 (0.02)**[*] |
| **Employment** | **-0.23 (0.001)**[**] |
| Educational level | -0.13 (0.07) |
| Marital status | 0.05 (0.43) |
| **Relationship with patient** | **0.15 (0.03)**[*] |
| **Living with patient** | **0.18 (0.009)**[**] |
| Having comorbidities | 0.09 (0.18) |
| Deliver care for another patient | 0.07 (0.32) |
| Time of caring in hours | 0.12 (0.09) |

[*] P value Significant on 0.05.

[**] P value Significant on 0.001.

**Table 5. Hierarchical multiple regression model for predictors of caregivers' burden.**

| | variables | Adjusted $R^2$ | SE | $R^2$ change | Standardized coefficient β | P | CI |
|---|---|---|---|---|---|---|---|
| Model 1 | age | 0.13 | 0.34 | 0.28 | -0.585 | **0.01** | -.02- -0.003 |
| | gender | | | | -0.014 | 0.79 | -0.18–0.24 |
| | employment | | | | -0.228 | 0.24 | -0.36–0.09 |
| | Relationship with patients | | | | 0.067 | 0.95 | -0.58–0.54 |
| | Education level | | | | -0.334 | **0.02** | -0.48- -0.05 |
| | Marital status | | | | -0.195 | 0.53 | -0.29- -0.16 |
| | Live with patient | | | | 0.091 | 0.92 | -0.25–0.22 |
| | Complain of chronic diseases | | | | 0.253 | 0.41 | -0.13–0.32 |
| | Deliver care for another patient | | | | -0.042 | 0.98 | -0.21–0.20 |
| | Time spent in caring of patient | | | | 0.123 | 0.2 | -0.05–0.23 |
| Model 2 | QLI | 0.32 | 0.29 | 0.17 | 0.446 | **0.001** | 0.04–0.13 |

SE: standard error.

CI: Confidence Interval.

QLI: Quality of life index.

## Discussion

### Caregiver burden

Our study found a moderate level of caregiver burden. The highest score on the caregiver burden scale in this study was related to personal care and motivational support. While the lowest was regarding emotional burden.

Most of the caregivers in the study were family members, particularly a spouse or the patient's children who usually lived with the patient. Close relatives have been recognized as "hidden" carers who have an increased risk of experiencing caregiver burden, but they do not recognize themselves as carers because they consider their caring role as a natural consequence of a relationship such as being a spouse [22]. This is consistent with Jordanian norms and culture, in which close relatives are obligated to take care of their loved ones as they acknowledge this mission as an extension of their roles and life responsibilities. This result is consistent with other studies, especially in Asian countries that have similar norms [10]. Another study found that not only the type of the relationship but also its quality affects the caregiver's level of strain [23].

A better understanding of caregiver burden is important to provide well planned support to help caregivers care for the patients with cardiac diseases in their lives. Informal caregivers play a vital role in increasing the QoL for patients prone to more frequent use of acute care services. The family is still recognized as a source of emotional and social support throughout the patient's journey of disease and recovery. Therefore, the level of burden in the current study was moderate, not high. The possible explanation is that family members share the burden of providing care in particulate when the affected person is one of the parents [24]. That is, sharing this responsibility reduce the burden on the caregivers. This is also similar to the tradition in Turkish society of accepting the caregiving role and perceiving this role as providing "help" rather than receiving a "burden" [24].

This moderate level was consistent with other studies, including a study by Yigitalp et al. [13] that found 20–30% of the partners perceived a moderate caregiver burden and therefore had a higher risk of poor mental health and decreased perceived control, while other studies found a mild burden among cancer patients' caregivers [15]. On the other hand, one study

found that the caregiver burden of those who cared for patients with HF was remarkably high [25].

The highest score on the caregiver burden scale in this study was related to personal care and motivational support. Similarly, a study by Luttik et al. [19] found that the burden of caregivers increased with personal care (different tasks), lack of family support, and when the caregiver was female. On the other hand, the lowest burden was on emotional status.

The "personal care and motivational support elements" of the care burden in this study can be explained by several factors, such as the close proximity of living with the patient and the long hours spent in providing care. The caregivers provided voluntary care to their family members, but, at the same time, they were not familiar with what should or should not be done because they did not have enough experience or knowledge regarding this care. In addition, there is limited training on how to carry out this role, which may cause uneasiness and stress for the caregivers [26].

Notably, this study showed a "limited emotional burden" among caregivers; however, this does not cancel out the fact that earlier research has emphasized the need for emotional support for caregivers as most of them are offering support because they are emotionally involved, especially when family members are the focus of the care [27]. Furthermore, others stressed the need for emotional control in a good quality relationship between the caregiver and the patient to gain better health outcomes for both parties [28].

It should be taken into consideration in this context that some cultural and religious variables that are characteristic of the studied population may have diminished the emotional element of the care burden.

## Caregiver burden and related sociodemographic

This study found that female caregiver, unemployment, not being a close relative to the cardiac patient, or living with the patient, will lead to higher burden and stress. Similarly, the caregiver burden increased for those who were unemployed, single, spending more hours with the patient, depressed, and having more physical health problems [29].

Also, the results were consistent with other studies which confirm that women are more often caregivers of patients with chronic illnesses compared to men, especially Jordanian women usually take care of the sick. Similarly previous research has shown that female caregivers experience higher levels of caregiver burden and depression and lower levels of physical health and well-being compared to their male counterparts [30, 31].

Similar results were observed in Adelman's study (2014) that found that females are demographically more susceptible to caregiver burden than males, living with the patient, low educational levels, social isolation, depression, financial stress, length of caregiving time, and the inescapability of caregiving duties [32]. All of these factors should be taken into consideration to reduce the caregiver burden.

## Predictors of caregiver burden

The results demonstrated that being young and with a low educational level increased the burden of the caregiver, also a high level of QoL of the cardiac patient increases the caregiver burden.

Being young with low educational level associated with lower financial support for paying the costs of drugs, rehabilitation, referrals, transportation, and bills. This will increase the burden of caregivers. These results are consistent with the other studies [33, 31].

The novel aspect of this study is the relationship between cardiac patients' QoL and the caregiver burden due to the limited number of studies on this subject. It is noteworthy that the

burden increased when the patients' QoL increased; this is consistent with the study by Bidwell et al. [23] in which they found that when the QoL of cardiac patients improved at one month post discharge after left ventricular assist device therapy, the caregiver burden was the highest. The same result was found in the meta-analysis study by Bidwell et al [34], which observed that improving the QoL of cardiac patients led to more physical and emotional strain on their caregivers and more time spent providing care for which they were unprepared, especially as most of them were young, female, and uneducated.

While the cross-sectional nature of the current study does not allow inferences to be drawn about the relationships between patient QoL and caregiver burden, we can speculate about possible explanations for the correlation. Day-to-day life with patients with cardiac disease often brings new physical, psychosocial, and financial challenges for families. To address these challenges caregivers may exert more efforts to keep their patients in good QoL, experiencing hardships and stress in the process [35]. This may be heightened by the threat of losing a partner and/or the added burden of caring for someone who was previously able to contribute to household responsibilities.

Another aspect that should be considered is that most of these caregivers live with these patients, which also increases the load on them, especially when a caregiver has to take on additional roles in their lives due to their partner's ill health. So the type of relationship is another important factor that future studies should investigate.

Characteristics associated with caregiver burden may have a predictive value in identifying those at risk of prolonged exposure to elevated stress and its associated impact on morbidity and mortality [36] Caregivers have been found to perceive the burden of patients' symptoms as being greater than the patients themselves perceive it [37]. This may demonstrate the strong relationship between caregivers and patients which may lead to an improvement in patients' QoL, so relationship quality is an important predictive factor that is recommended to be tested repeatedly in future studies.

This study had many strengths such as the novelty of studying QoL of cardiac patients correlated with caregiver burden. Furthermore, many sociodemographic factors collected from the literature were studied to explore their effect on caregiver burden. Finally, the study had a good sample size. This does not mean that there are no limitations that should be taken into consideration, including the convenience sampling. Using of a cross-sectional design, limits causal inferences about the nature and direction of the relationships observed, so generalization is limited.

## Conclusion

This study found a correlation between patient QOL and caregiver burden. Caregivers should be offered more support by healthcare providers, including training on the required skills or practices needed in providing care for their cardiac patients, to reduce the strain of caregivers as it was found in this study that the burden was higher due to a lack of personal care skills. Future studies are recommended to investigate the effect of the relationship type and quality between caregivers and their patients which was found to have a strong influence, and to include socioeconomic status. A longitudinal study to measure the caregiver burden is necessary.

## Acknowledgments

Authors acknowledge the highly appreciated guidance offered by the teacher Reem Jarrad from The Nursing School in The University of Jordan, and Dr Omar Al Omari from Sultan Qaboos University for their kind review and valuable comments.

## Author Contributions

**Conceptualization:** Maha Subih, Marwa AlBarmawi, Dalal Yehia Bashir, Shrooq Munir Jacoub.

**Data curation:** Maha Subih, Najah Sayyah Sayyah.

**Formal analysis:** Maha Subih, Marwa AlBarmawi.

**Funding acquisition:** Maha Subih, Dalal Yehia Bashir.

**Methodology:** Maha Subih, Dalal Yehia Bashir, Shrooq Munir Jacoub.

**Project administration:** Maha Subih.

**Resources:** Najah Sayyah Sayyah.

**Supervision:** Maha Subih.

**Validation:** Maha Subih, Marwa AlBarmawi, Dalal Yehia Bashir, Shrooq Munir Jacoub, Najah Sayyah Sayyah.

**Writing – original draft:** Maha Subih, Najah Sayyah Sayyah.

**Writing – review & editing:** Maha Subih, Marwa AlBarmawi, Dalal Yehia Bashir, Shrooq Munir Jacoub.

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
