## [Decision Letter · Decision Letter 0]

26 May 2020

PONE-D-20-10353

Quality of life of Cardiac patients as a predictor of Caregivers’ Burden

PLOS ONE

Dear Dr. Subih,

Thank you for submitting your manuscript to PLOS ONE. After careful consideration, we feel that it has merit but does not fully meet PLOS ONE’s publication criteria as it currently stands. Therefore, we invite you to submit a revised version of the manuscript that addresses the points raised during the review process.

We look forward to receiving your revised manuscript.

Kind regards,

Tim Luckett

Academic Editor

PLOS ONE

Journal Requirements:

'This work was supported by Al Zaytoonah University of Jordan (ZUJ) (Grant number

21/172/2018)'

'MS, MA, DY, SJ, NS

750$ to each one

Al Zaytoonah University of Jordan

www.zuj.edu.jo

no any role in the study'

Additional Editor Comments (if provided):

Reviewers' comments:

Reviewer's Responses to Questions

**Comments to the Author**

1. Is the manuscript technically sound, and do the data support the conclusions?

Reviewer #1: Yes

Reviewer #2: No

2. Has the statistical analysis been performed appropriately and rigorously? 

Reviewer #1: Yes

Reviewer #2: No

3. Have the authors made all data underlying the findings in their manuscript fully available?

Reviewer #1: Yes

Reviewer #2: Yes

4. Is the manuscript presented in an intelligible fashion and written in standard English?

Reviewer #1: Yes

Reviewer #2: No

5. Review Comments to the Author

Reviewer #1: The aim of the study was to determine predictors of caregiver burden. The authors examined 200 cardiac patients and their caregivers. Identified predictors of caregiver burden were age, educational level and HRQoL.

Of note the authors translated the DOB instrument and QLI-Cardiac 4 instrument to Arabic and tested its internal consistency and construct validity.

Mrthods:

‘the calculated sample size was 180 caregivers at α = 0.05 and power 83 = 0.8.’

The authors should indicate to what studied variable does this power calculation apply to.

Results:

‘Most cardiac patients were diagnosed as post angioplasty 142 and STENT (32%)’.

This statement is not factually accurate since 32% is less than half so it can not be designated as most. It seems that most patients had HF, not stent.

The DOBI results seem to be one of the main results. Most readers will not know what the scores exactly mean. The authors should describe the details with some more detail helping the reader understand the findings.

The abstract conclusion states that ‘Issues concerning the increased level of burden among caregivers is related to some modifiable factors’. Since age and educational level are not really modifiable, the conclusion should be revised. Maybe along the lines what was porposed in the discussion, that ‘caregivers may benefit from some targeted educational and training programmes to develop their coping and resilience skills in a way that decreases their care burden and improves their quality of care and self-confidence’. To tie it up with the results, maybe the best way would be to propose that patients whose caregivers are younger and have lower educational level are at the highest risk of elevated caregiver burden and targeted interventions for this group of caregivers should be tested to assist them.

Reviewer #2: Comments to Author:

Ref. No.: PONE-D-20-10353

The work by Subih et al. entitled ”Quality of life of Cardiac patients as a predictor of Caregivers’ Burden” focuses on the association between caregivers burden and the quality of life of the cardiac patients they care for – investigating the caring of carers. The authors report that lower age, shorter education and higher quality of life of the patients were all statistically significant predictors of the caregivers burden. The objective of the study is interesting, the manuscript is engaging and the methodology is broadly appropriate (which is often challenging within studies focusing on Quality of Life). However, I have some concern associated with the structure on the manuscript, the language used and reporting of data listed below.

MAJOR

1. For a broader audience in a journal as PLOS One please specify concepts and the setting early in the introduction, as they are somewhat unknown for the reader. I suggest a re-structuring of the introduction. E.g. move page 3 line 53-55 up – as it contains key background information.

2. I find some of the description regarding the method section too detailed while the description of some important methods are not adequately described.

3. The study assessed 200 caregivers and 203 patients, however the response rate around 80%. There are some figures e.g. in table 1 that do not quite align with this. E.g. the title n=200, but you previously stated a response rate of 80%? Furthermore, the population (in this cross-sectional design) stems from “one educational, two public, and three private hospital clinics in Jordan”. With the inherent limitations of a cross-sectional design regarding selection bias and generalizability I would suggest a more rigorous description of how the population was chosen.

4. In the conclusion you state that “Therefore, we recommend that future interventional studies to test such training programmes evaluate their efficiency levels in terms of outcomes for both patients and their caregivers in the long term.” but the objective of the study was to “explore the predictors of caregivers´ burden.” I suggest a more stringent focus on concluding on the findings of the study. Are there evidence in current training programs?

5. The data is interesting, however their presentation needs adjustment. According to table 2-4 I would appreciate if you reported the DOBI and QLI-Cardiac 4 findings in context with the demographic variables. E.g. is there a relationship between the total DOBI scale and the time spent with the patient? Would it be more appropriate to add a figure for this purpose? It would be very illustrative with a forest plot.

6. The study assesses the caregivers burden by the DOBI in a Arabic version. Please reference that this translated version is validated. I am not aware, whether the approach with submitting the instrument to local faculty members for approval is a standardized method. Please clarify.

7. Please discuss on the finding that the caregiving burden increases when the patients HRQoL increases. A potential cause?

8. The title of the manuscript is not completely aligned with the conclusion in the abstract. I would suggest adjusting.

9. Please consider grammatical assistance or make adjustments in the language throughout.

10. In the abstract conclusion you describe that the burden “is related to some modifiable factors” and state that the variables that are of significance are age, years of educational level and quality of life of the patients. Age is hardly modifiable, years of educational level might be, however you find that increased quality of life of the patients is associated with a higher burden. Its seems counterintuitive to modify the latter, please elaborate.

MINOR

1. Do you have more data on the socioeconomic position? It would be interesting to know and potentially be a relevant confounding variable regarding economy?

2. Have you gathered data on the quality of the relationship between caregiver and the patient? Literature suggests that this is highly relevant.

3. Why was a cardiac population chosen? Please clarify.

4. In the Health-related quality of life is mentioned whereas Quality of Life is used without. Please elaborate on your definition on the difference.

5. Please detail the data analysis section on the descriptive statistics.

6. Please specify which correlation tests were used in the data analysis section.

7. Page 2 line 35: “Improving patient´s outcomes is now the focus?

8. Page 2 linje 40: You have just abbreviated HRQoL, please use abbreviation.

9. Please reference statement “The health-related quality of life for cardiac patients and the burden of care to the caregivers is core for healthcare planning, especially when no previous or limited experience in these conditions exists.”

10. Page 3: Line 43-45: Please re-write as the current version comes across as obvious.

11. Page 3 line 45: Use a different wording than “somewhat mandatory”. Please specify.

12. Page 3 line 54. Is the use of family members as providers of care to patients unique for the Arab and Islamic culture or similar across geography and cultures? Please specify for the broad audience. Please put the findings into a broader context e.g. western culture.

13. Page 4 line 70: “Characteristics of those caregivers”. There seems to be a missing a part of the sentence? “those caregivers that….”?

14. Page 4 line 71 “…chilvarous…” I suggest finding a more appropriate academic wording? Although I agree on the purpose of the wording.

15. Page 4 line 82: Please specify the statistical software G power, by version used and country of origin.

16. The study uses a convenience sample as population. Please add this to the limitations section discussing potential sampling error and lack of achieving a representative population.

17. Page 4 line 91: “…would they like to take part in the research” please rephrase.

18. Page 5 line 94-95. “with the option to withdraw at any time, even if they had already started to participate”. Please erase second part of the sentence.

19. Page 5 line 97 “of our university” Please write the specific university.

20. Page 6 line 135 “Two hundred caregivers completed the questionnaire (response rate of 80%).” It is not clear to me what the actual number is based on this sentence. Was it two hundred completed (please use actual numbers) or 200*0,8?

21. During the results section please use mean/ median values rather than “average”.

22. Page 7 line 142. Please specify what you mean by a diagnosis of stent?

23. Please erase sentence “Mean scores on the DOBI…” as this is correctly states in the method section

24. I suggest beginning the discussion with stating your main findings.

25. Page 10 Line 214-215 “When love and beliefs synergise to cover the act of giving with the umbrealla of the blessing of a higher power, some psychological factors may be modified” Please add reference for academic rigorous.

26. Page 10 line 237 Please use different verb than conduct in relation to generalization.

27. In table 1. Please specify what “other” entail.

28. In table 1. I suggest using n (%) instead of % (n) version.

29. In table 1 you find that the mean time spent in caring is 6,9 hours, which seems remarkably high if it is direct contact hours with the patient (around 2/3 of the awake hours of the caregiver). Please clarify on this.

30. In table 1. You state “Type of condition” as a variable. I would term “HF” a condition, but not necessarily the remaining. PTCA is not a condition. Post-surgery is on the other hand. Please clarify your choice.

31. In table 1. Add abbreviation to CABG.

32. In table 4. Please report all the demographic variables and their association to the caregivers burden instead of solely those of statistical significance.

33. Table 4. I would suggest a more standardized format of presenting your findings.

34. It is not stated whether the study used complete case analysis or imputed for missing data – and if which statistical imputation was used.

35. I would suggest reporting mean scores with SD instead of just means.

36. The terms prediction and association seems to be loosely used, however I suggest the usage of association instead of prediction throughout

37. Remember to include the strengths of the study and not only report your limitations.

38. In introduction it is stated “Improving patients’ outcomes is now the focus of national and international healthcare plans, especially among nurse practitioners (2)”, however your referenced article is from 20012. Have no new articles on this matter been published?

6. PLOS authors have the option to publish the peer review history of their article (what does this mean?). If published, this will include your full peer review and any attached files.

Reviewer #1: Yes: Josef Stehlik

Reviewer #2: No

---

## [Author Response · Author response to Decision Letter 0]

27 Jun 2020

response to reviewers had been added, also manuscript with revised track change and clean manuscript added.

in cover letter we added fund section from Al Zaytoonah University with the approval number

No laboratory protocols

funding statement removed from the manuscript (funding statement declared)

no ethical or legal restrictions on data

conclusion as the whole manuscript had been changed and revised (editing and rewritten in standard English language by professional company done again)

all statistical analysis comments had been taken in consideration and corrected

power of sample size was indicated for using regression (10 predictors) which had clarified in the revised manuscript 

STENT had been changed to post PCI(percutanous coronary intervention) which is more specific and scientific

most cases were adjusted as reviewers said (which were HF patients)

Abstract had modified especially conclusion as reviewers asked

all major and minor changes that reviewers requested had been done

---

## [Decision Letter · Decision Letter 1]

3 Jul 2020

PONE-D-20-10353R1

Quality of life of Cardiac patients as a predictor of Caregivers’ Burden

PLOS ONE

Dear Dr. Subih,

Thank you for submitting your manuscript to PLOS ONE. After careful consideration, we feel that it has merit but does not fully meet PLOS ONE’s publication criteria as it currently stands. Therefore, we invite you to submit a revised version of the manuscript that addresses the points raised during the review process.

We look forward to receiving your revised manuscript.

Kind regards,

Tim Luckett

Academic Editor

PLOS ONE

Additional Editor Comments (if provided):

Given this is a cross-sectional study, I think the term 'predict/predictor/prediction' should be replaced with 'correlation' or 'association' throughout, including the title.

Intro and objectives -

Please delete the sentence at the end of the Introduction, which is repetitious and poorly constructed: 'Hence the novelty of this study is to examine the QoL of cardiac patients with the caregiver burden'.

Please rephrase the objectives to read: 'The purpose of this paper was to assess the QoL of cardiac patients and its association with caregivers’ burden alongside other variables'.

Methods -

Please remove reference to anonymity as, technically, the data were re-identifiable.

Please change to: 'Cronbach’s alpha reliability was previously found to be between 0.80 and 0.85'.

Please change to: "Cronbach’s alphas have previously measured between .84 and .98 for the original QLI-Cardiac 4 (total scale)'

Please change data to be plural rather than singular throughout (e.g. 'were' not 'was' analysed).

Discussion -

Please change 'our norms and culture'' to 'Jordanian norms and culture'.

Please explain further the following sentence: 'Therefore, the caregiver burden was moderate in this study as the burden of providing care for the patient after discharge primarily lies with family members, especially if the patient with cardiac disease is a parent'

New paragraph commencing with 'Day-to-day life with patients with cardiac disease ...' - does the first sentence imply that there may be a causal relationship between caregiver burden and patient QOL? If so, given this is cross-sectional data, an alternative interpretation might also need to be offered that sees the relationship the other way around or merely associative and not causal. I don't see how the other sentences in this paragraph refer to the correlation between patient QOL and CG burden?

The limitation on convenience sampling and generalisability should be separated from the one on cross-sectional data and causal inference in two different sentences.

Conclusion -

Please remove the first sentence which does not relate to findings from the study and replace with one that restates that this study found a correlation between patient QOL and caregiver burden.

Reviewers' comments:

Reviewer's Responses to Questions

**Comments to the Author**

1. If the authors have adequately addressed your comments raised in a previous round of review and you feel that this manuscript is now acceptable for publication, you may indicate that here to bypass the “Comments to the Author” section, enter your conflict of interest statement in the “Confidential to Editor” section, and submit your "Accept" recommendation.

Reviewer #1: All comments have been addressed

2. Is the manuscript technically sound, and do the data support the conclusions?

Reviewer #1: Yes

3. Has the statistical analysis been performed appropriately and rigorously? 

Reviewer #1: Yes

4. Have the authors made all data underlying the findings in their manuscript fully available?

Reviewer #1: Yes

5. Is the manuscript presented in an intelligible fashion and written in standard English?

Reviewer #1: Yes

6. Review Comments to the Author

Reviewer #1: The authors were responsive to the reviewers.

The queries have been addressed.

The syntax of the paper has been improved.

7. PLOS authors have the option to publish the peer review history of their article (what does this mean?). If published, this will include your full peer review and any attached files.

Reviewer #1: No

---

## [Author Response · Author response to Decision Letter 1]

9 Jul 2020

all editor requests had been taken in consideration and modifed

---

## [Editor Report · Decision Letter 2]

16 Jul 2020

PONE-D-20-10353R2

Correlation between quality of life of cardiac patients and caregiver burden

PLOS ONE

Dear Dr. Subih,

Thank you for submitting your manuscript to PLOS ONE. After careful consideration, we feel that it has merit but does not fully meet PLOS ONE’s publication criteria as it currently stands. Therefore, we invite you to submit a revised version of the manuscript that addresses the points raised during the review process.

Specifically, the insertion of "the relationship between the QoL and caregiver burden are going in both directions" into the Discussion cannot be justified based on the available data, nor does it adequately introduce the remainder of the paragraph, which seems to only focus on one direction and also needs editing for English. I suggest replacing this paragraphwith the following: "While the cross-sectional nature of the current study does not allow inferences to be drawn about the relationships between patient QoL and caregiver burden, we can speculate about possible explanations for the correlation. Day-to-day life with patients with cardiac disease often brings new physical, psychosocial, and financial challenges for families. To address these challenges caregivers may exert more efforts to keep their patients in good QoL, experiencing hardships and stress in the process (35). This may be heightened by the threat of losing a partner and/or the added burden of caring for someone who was previously able to contribute to household responsibilities."

We look forward to receiving your revised manuscript.

Kind regards,

Tim Luckett

Academic Editor

PLOS ONE

---

## [Author Response · Author response to Decision Letter 2]

17 Jul 2020

the paragraph that the editor request to change had been changed.

---

## [Editor Report · Decision Letter 3]

21 Jul 2020

Correlation between quality of life of cardiac patients and caregiver burden

PONE-D-20-10353R3

Dear Dr. Subih,

We’re pleased to inform you that your manuscript has been judged scientifically suitable for publication and will be formally accepted for publication once it meets all outstanding technical requirements.

Kind regards,

Tim Luckett

Academic Editor

PLOS ONE
---

## [Editor Report · Acceptance letter]

22 Jul 2020

PONE-D-20-10353R3 

Correlation between quality of life of cardiac patients and caregiver burden 

Dear Dr. Subih:

I'm pleased to inform you that your manuscript has been deemed suitable for publication in PLOS ONE. Congratulations! Your manuscript is now with our production department. 

Kind regards, 

on behalf of

Dr. Tim Luckett 

Academic Editor

PLOS ONE